# Unsupervised Translation of Programming Languages

**Baptiste Roziere**[*]
Facebook AI Research
Paris-Dauphine University
broz@fb.com

**Marie-Anne Lachaux**[*]
Facebook AI Research
malachaux@fb.com

**Lowik Chanussot**
Facebook AI Research
lowik@fb.com

**Guillaume Lample**
Facebook AI Research
glample@fb.com

## Abstract

A transcompiler, also known as source-to-source translator, is a system that converts source code from a high-level programming language (such as C++ or Python) to another. Transcompilers are primarily used for interoperability, and to port codebases written in an obsolete or deprecated language (e.g. COBOL, Python 2) to a modern one. They typically rely on handcrafted rewrite rules, applied to the source code abstract syntax tree. Unfortunately, the resulting translations often lack readability, fail to respect the target language conventions, and require manual modifications in order to work properly. The overall translation process is time-consuming and requires expertise in both the source and target languages, making code-translation projects expensive. Although neural models significantly outperform their rule-based counterparts in the context of natural language translation, their applications to transcompilation have been limited due to the scarcity of parallel data in this domain. In this paper, we propose to leverage recent approaches in unsupervised machine translation to train a fully unsupervised neural transcompiler. We train our model on source code from open source GitHub projects, and show that it can translate functions between C++, Java, and Python with high accuracy. Our method relies exclusively on monolingual source code, requires no expertise in the source or target languages, and can easily be generalized to other programming languages. We also build and release a test set composed of 852 parallel functions, along with unit tests to check the correctness of translations. We show that our model outperforms rule-based commercial baselines by a significant margin.

## 1 Introduction

A transcompiler, transpiler, or source-to-source compiler, is a translator which converts between programming languages that operate at a similar level of abstraction. Transcompilers differ from traditional compilers that translate source code from a high-level to a lower-level programming language (e.g. assembly language) to create an executable. Initially, transcompilers were developed to port source code between different platforms (e.g. convert source code designed for the Intel 8080 processor to make it compatible with the Intel 8086). More recently, new languages have been developed (e.g. CoffeeScript, TypeScript, Dart, Haxe) along with dedicated transcompilers that convert them into a popular or omnipresent language (e.g. JavaScript). These new languages address some shortcomings of the target language by providing new features such as list comprehension (CoffeeScript), object-oriented programming and type checking (TypeScript), while detecting errors and providing optimizations. Unlike traditional programming languages, these new languages are

---

[*]Equal contribution. The order was determined randomly.

designed to be translated with a perfect accuracy (i.e. the compiled language does not require manual adjustments to work properly). In this paper, we are more interested in the traditional type of transcompilers, where typical use cases are to translate an existing codebase written in an obsolete or deprecated language (e.g. COBOL, Python 2) to a recent one, or to integrate code written in a different language to an existing codebase.

Migrating an existing codebase to a modern or more efficient language like Java or C++ requires expertise in both the source and target languages, and is often costly. For instance, the Commonwealth Bank of Australia spent around $750 million and 5 years of work to convert its platform from COBOL to Java. Using a transcompiler and manually adjusting the output source code may be a faster and cheaper solution than rewriting the entire codebase from scratch. In natural language, recent advances in neural machine translation have been widely accepted, even among professional translators, who rely more and more on automated machine translation systems. A similar phenomenon could be observed in programming language translation in the future.

Translating source code from one Turing-complete language to another is always possible in theory. Unfortunately, building a translator is difficult in practice: different languages can have a different syntax and rely on different platform APIs and standard-library functions. Currently, the majority of transcompilation tools are rule-based; they essentially tokenize the input source code and convert it into an Abstract Syntax Tree (AST) on which they apply handcrafted rewrite rules. Creating them requires a lot of time, and advanced knowledge in both the source and target languages. Moreover, translating from a dynamically-typed language (e.g. Python) to a statically-typed language (e.g. Java) requires to infer the variable types which is difficult (and not always possible) in itself.

The applications of neural machine translation (NMT) to programming languages have been limited so far, mainly because of the lack of parallel resources available in this domain. In this paper, we propose to apply recent approaches in unsupervised machine translation, by leveraging large amount of monolingual source code from GitHub to train a model, TransCoder, to translate between three popular languages: C++, Java and Python. To evaluate our model, we create a test set of 852 parallel functions, along with associated unit tests. Although never provided with parallel data, the model manages to translate functions with a high accuracy, and to properly align functions from the standard library across the three languages, outperforming rule-based and commercial baselines by a significant margin. Our approach is simple, does not require any expertise in the source or target languages, and can easily be extended to most programming languages. Although not perfect, the model could help to reduce the amount of work and the level of expertise required to successfully translate a codebase. The main contributions of the paper are the following:

- We introduce a new approach to translate functions from a programming language to another, that is purely based on monolingual source code.
- We show that TransCoder successfully manages to grasp complex patterns specific to each language, and to translate them to other languages.
- We show that a fully unsupervised method can outperform commercial systems that leverage rule-based methods and advanced programming knowledge.
- We build and release a validation and a test set composed of 852 parallel functions in 3 languages, along with unit tests to evaluate the correctness of generated translations.
- We will make our code and pretrained models publicly available.

## 2   Related work

**Source-to-source translation.**    Several studies have investigated the possibility to translate programming languages with machine translation. For instance, Nguyen et al. [36] trained a Phrase-Based Statistical Machine Translation (PBSMT) model, Moses [27], on a Java-C# parallel corpus. They created their dataset using the implementations of two open source projects, Lucene and db4o, developed in Java and ported to C#. Similarly, Karaivanov et al. [22] developed a tool to mine parallel datasets from ported open source projects. Aggarwal et al. [1] trained Moses on a Python 2 to Python 3 parallel corpus created with 2to3, a Python library [2] developed to port Python 2 code to Python 3. Chen et al. [12] used the Java-C# dataset of Nguyen et al. [36] to translate code with tree-to-tree neural networks.

They also use a transcompiler to create a parallel dataset CoffeeScript-Javascript. Unfortunately, all these approaches are supervised, and rely either on the existence of open source projects available in multiple languages, or on existing transcompilers, to create parallel data. Moreover, they essentially rely on BLEU score [38] to evaluate their translations [1, 10, 22, 36], which is not a reliable metric, as a generation can be a valid translation while being very different from the reference.

**Translating from source code.** Other studies have investigated the use of machine translation from source code. For instance, Oda et al. [37] trained a PBSMT model to generate pseudo-code. To create a training set, they hired programmers to write the pseudo-code of existing Python functions. Barone and Sennrich [10] built a corpus of Python functions with their docstrings from open source GitHub repositories. They showed that a neural machine translation model could be used to map functions to their associated docstrings, and vice versa. Similarly, Hu et al. [21] proposed a neural approach, DeepCom, to automatically generate code comments for Java methods.

**Other applications.** Another line of work studied the applications of neural networks to code suggestion [2, 11, 34], or error detection [13, 18, 47]. Recent approaches have also investigated the use of neural approaches for code decompilation [16, 24]. For instance, Katz et al. [23] propose a sequence-to-sequence model to predict the C code of binary programs. A common issue with standard seq2seq models, is that the generated functions are not guaranteed to compile, and even to be syntactically correct. To address this issue, several approaches proposed to use additional constraints on the decoder, to ensure that the generated functions respect the syntax of the target language [3, 4, 5, 40, 48]. Recently, Feng et al. [15] introduced Codebert, a transformer pretrained with a BERT-like objective [14] on open source GitHub repositories. They showed that pretraining improves the performance on several downstream tasks such as code documentation generation and code completion.

**Unsupervised Machine Translation.** The quality of NMT systems highly depends on the quality of the available parallel data. However, for the majority of languages, parallel resources are rare or nonexistent. Since creating parallel corpora for training is not realistic (creating a small parallel corpus for evaluation is already challenging [19]), some approaches have investigated the use of monolingual data to improve existing machine translation systems [17, 20, 41, 49]. More recently, several methods were proposed to train a machine translation system exclusively from monolingual corpora, using either neural models [30, 8] and statistical models [32, 7]. We describe now some of these methods and how they can be instantiated in the setting of unsupervised transcompilation.

## 3 Model

For TransCoder, we consider a sequence-to-sequence (seq2seq) model with attention [44, 9], composed of an encoder and a decoder with a transformer architecture [45]. We use a single shared model for all programming languages. We train it using the three principles of unsupervised machine translation identified in Lample et al. [32], namely initialization, language modeling, and back-translation. In this section, we summarize these principles and detail how we instantiate them to translate programming languages. An illustration of our approach is given in Figure 1.

### 3.1 Cross Programming Language Model pretraining

Pretraining is a key ingredient of unsupervised machine translation Lample et al. [32]. It ensures that sequences with a similar meaning are mapped to the same latent representation, regardless of their languages. Originally, pretraining was done by initializing the model with cross-lingual word representations [30, 8]. In the context of unsupervised English-French translation, the embedding of the word "cat" will be close to the embedding of its French translation "chat". Cross-lingual word embeddings can be obtained by training monolingual word embeddings and aligning them in an unsupervised manner [31, 6].

Subsequent work showed that pretraining the entire model (and not only word representations) in a cross-lingual way could lead to significant improvements in unsupervised machine translation [29, 33, 43]. In particular, we follow the pretraining strategy of Lample and Conneau [29], where a Cross-lingual Language Model (XLM) is pretrained with a masked language modeling objective [14] on monolingual source code datasets.

Cross-lingual Masked Language Model pretraining

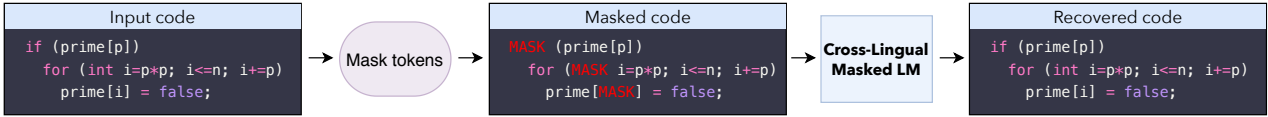

Denoising auto-encoding

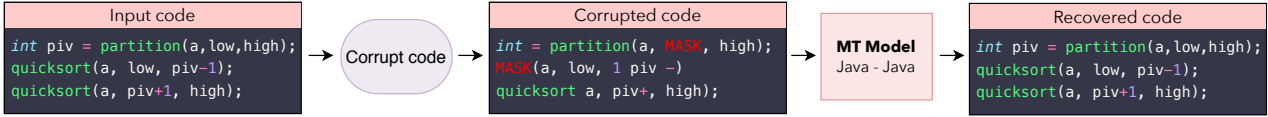

Back-translation

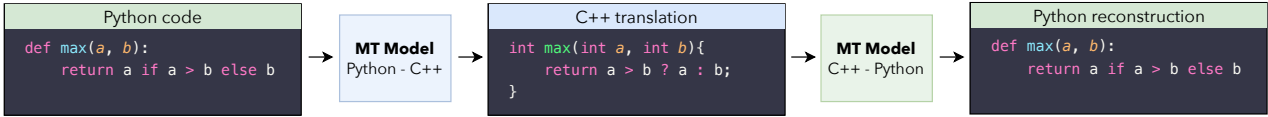

Figure 1: **Illustration of the three principles of unsupervised machine translation used by our approach.** The first principle initializes the model with cross-lingual masked language model pretraining. As a result, pieces of code that express the same instructions are mapped to the same representation, regardless of the programming language. Denoising auto-encoding, the second principle, trains the decoder to always generate valid sequences, even when fed with noisy data, and increases the encoder robustness to input noise. Back-translation, the last principle, allows the model to generate parallel data which can be used for training. Whenever the Python → C++ model becomes better, it generates more accurate data for the C++ → Python model, and vice versa. Figure 5 in the appendix provides a representation of the cross-lingual embeddings we obtain after training.

The cross-lingual nature of the resulting model comes from the significant number of common tokens (anchor points) that exist across languages. In the context of English-French translation, the anchor points consists essentially of digits and city and people names. In programming languages, these anchor points come from common keywords (e.g. for, while, if, try), and also digits, mathematical operators, and English strings that appear in the source code. [3]

For the masked language modeling (MLM) objective, at each iteration we consider an input stream of source code sequences, randomly mask out some of the tokens, and train TransCoder to predict the tokens that have been masked out based on their contexts. We alternate between streams of batches of different languages. This allows the model to create high quality, cross-lingual sequence representations. An example of XLM pretraining is given on top of Figure 1.

## 3.2 Denoising auto-encoding

We initialize the encoder and decoder of the seq2seq model with the XLM model pretrained in Section 3.1. The initialization is straightforward for the encoder, as it has the same architecture as the XLM model. The transformer decoder, however, has extra parameters related to the source attention mechanism [45]. Following Lample and Conneau [29], we initialize these parameters randomly.

XLM pretraining allows the seq2seq model to generate high quality representations of input sequences. However, the decoder lacks the capacity to translate, as it has never been trained to decode a sequence based on a source representation. To address this issue, we train the model to encode and decode sequences with a Denoising Auto-Encoding (DAE) objective [46]. The DAE objective operates like a supervised machine translation algorithm, where the model is trained to predict a sequence of tokens given a corrupted version of that sequence. To corrupt a sequence, we use the same noise model as the one described in Lample et al. [30]. Namely, we randomly mask, remove and shuffle input tokens.

The first symbol given as input to the decoder is a special token indicating the output programming language. At test time, a Python sequence can be encoded by the model, and decoded using the C++ start symbol to generate a C++ translation. The quality of the C++ translation will depend on the "cross-linguality" of the model: if the Python function and a valid C++ translation are mapped to the same latent representation by the encoder, the decoder will successfully generate this C++ translation.

The DAE objective also trains the "language modeling" aspect of the model, i.e. the decoder is always trained to generate a valid function, even when the encoder output is noisy. Moreover it also trains the encoder to be robust to input noise, which is helpful in the context of back-translation where the model is trained with noisy input sequences. DAE is illustrated in the middle of Figure 1.

### 3.3 Back-translation

In practice, XLM pretraining and denoising auto-encoding alone are enough to generate translations. However, the quality of these translations tends to be low, as the model is never trained to do what it is expected to do at test time, i.e. to translate functions from one language to another. To address this issue, we use back-translation, which is one of the most effective methods to leverage monolingual data in a weakly-supervised scenario. Initially introduced to improve the performance of machine translation in the supervised setting [41], back-translation turned out to be an important component of unsupervised machine translation [30, 32, 8].

In the unsupervised setting, a source-to-target model is coupled with a backward target-to-source model trained in parallel. The target-to-source model is used to translate target sequences into the source language, producing noisy source sequences corresponding to the ground truth target sequences. The source-to-target model is then trained in a weakly supervised manner to reconstruct the target sequences from the noisy source sequences generated by the target-to-source model, and vice versa. The two models are trained in parallel until convergence. An example of back-translation is illustrated in Figure 1.

## 4 Experiments

### 4.1 Training details

We use a transformer with 6 layers, 8 attention heads, and set the dimensionality of the model to 1024. We use a single encoder and a single decoder for all programming languages. During XLM pretraining, we alternate between batches of C++, Java, and Python, composed of 32 sequences of source code of 512 tokens. At training time, we alternate between the denoising auto-encoding and back-translation objectives, and use batches of around 6000 tokens. We optimize TransCoder with the Adam optimizer [25], a learning rate of $10^{-4}$, and use the same learning rate scheduler as Vaswani et al. [45]. We implement our models in PyTorch [39] and train them on 32 V100 GPUs. We use float16 operations to speed up training and to reduce the memory usage of our models.

### 4.2 Training data

We download the GitHub public dataset available on Google BigQuery[4]. It contains more than 2.8 million open source GitHub repositories. We filter projects whose license explicitly permits the re-distribution of parts of the project, and select the C++, Java, and Python files within those projects. Ideally, a transcompiler should be able to translate whole projects. In this work, we decide to translate at function level. Unlike files or classes, functions are short enough to fit into a single batch, and working at function level allows for a simpler evaluation of the model with unit tests (c.f. Section 4.4). We pretrain TransCoder on all source code available, and train the denoising auto-encoding and back-translation objectives on functions only. Please refer to Section A.3 and Table 3 in the appendix for more details on how the functions are extracted, and for statistics about our training set. We carry out an ablation study to determine whether it is better to keep or remove comments from source code. Keeping comments in the source code increases the number of anchor points across languages, which results in a better overall performance (c.f. Table 6 in the appendix). Therefore, we keep them in our final datasets and experiments.

### 4.3  Preprocessing

Recent approaches in multilingual natural language processing tend to use a common tokenizer [28], and a shared vocabulary for all languages. This reduces the overall vocabulary size, and maximizes the token overlap between languages, improving the cross-linguality of the model [14, 29]. In our case, a universal tokenizer would be suboptimal, as different languages use different patterns and keywords. The logical operators && and || exist in C++ where they should be tokenized as a single token, but not in Python. The indentations are critical in Python as they define the code structure, but have no meaning in languages like C++ or Java. We use the `javalang`[5] tokenizer for Java, the tokenizer of the standard library for Python[6], and the `clang`[7] tokenizer for C++. These tokenizers ensure that meaningless modifications in the code (e.g. adding extra new lines or spaces) do not have any impact on the tokenized sequence. An example of tokenized code is given in Figure 3 in the appendix. We learn BPE codes [42] on extracted tokens, and split tokens into subword units. The BPE codes are learned with fastBPE[8] on the concatenation of tokenized C++, Java, and Python files.

### 4.4  Evaluation

GeeksforGeeks is an online platform[9] with computer science and programming articles. It gathers many coding problems and presents solutions in several programming languages. From these solutions, we extract a set of parallel functions in C++, Java, and Python, to create our validation and test sets. These functions not only return the same output, but also compute the result with similar algorithm. In Figure 4 in the appendix, we show an example of C++-Java-Python parallel function that determines whether an integer represented by a string is divisible by 13.

The majority of studies in source code translation use the **BLEU** score to evaluate the quality of generated functions [1, 10, 22, 36], or other metrics based on the relative overlap between the tokens in the translation and in the reference. A simple metric is to compute the **reference match**, i.e. the percentage of translations that perfectly match the ground truth reference [12]. A limitation of these metrics is that they do not take into account the syntactic correctness of the generations. Two programs with small syntactic discrepancies will have a high BLEU score while they could lead to very different compilation and computation outputs. Conversely, semantically equivalent programs with different implementations will have low BLEU scores. Instead, we introduce a new metric, the **computational accuracy**, that evaluates whether the hypothesis function generates the same outputs as the reference when given the same inputs. We consider that the hypothesis is correct if it gives the same output as the reference for every input. Section B and Table 4 in the appendix present more details on how we create these unit tests, and give statistics about our validation and test sets.

At inference, TransCoder can generate multiple translations using beam search decoding [26]. In machine translation, the considered hypotheses are typically the ones with the highest log-probabilities in the beam. In our case, we have access to unit tests to verify the correctness of the generated hypotheses, so we report two sets of results for our computational accuracy metric: **CA@N**, the percentage of functions with at least one correct translation in the beam, and **N Beams - CA@1** the percentage of functions where the hypothesis in the beam with the highest log-probability is a correct translation. We select our best model using greedy decoding (CA@1) for speed efficiency.

### 4.5  Results

We report the results on our test set in Table 1, using greedy decoding (beam size 1), for the three metrics presented in Section 4.4. In Table 2, we report our results with beam search decoding, and compare TransCoder to existing baselines. We give an example of unsupervised translation from Python to C++ in Figure 2.

**Evaluation metric differences.**  In Table 1, we observe that a very large fraction of translations differ from the reference, and are considered as invalid by the reference match metric although they

| Python input | Unsupervised C++ translation |
|---|---|

```python
def SumOfKsubArray(arr, n, k):
  Sum = 0
  S = deque()
  G = deque()
  for i in range(k):
    while (len(S) > 0 and arr[S[-1]] >= arr[i]):
      S.pop()
    while (len(G) > 0 and arr[G[-1]] <= arr[i]):
      G.pop()
    G.append(i)
    S.append(i)
  for i in range(k, n):
    Sum += arr[S[0]] + arr[G[0]]
    while (len(S) > 0 and S[0] <= i - k):
      S.popleft()
    while (len(G) > 0 and G[0] <= i - k):
      G.popleft()
    while (len(S) > 0 and arr[S[-1]] >= arr[i]):
      S.pop()
    while (len(G) > 0 and arr[G[-1]] <= arr[i]):
      G.pop()
    G.append(i)
    S.append(i)
  Sum += arr[S[0]] + arr[G[0]]
  return Sum
```

```cpp
int SumOfKsubArray(int arr[], int n, int k){
  int Sum = 0;
  deque <int> S;
  deque <int> G;
  for(int i = 0; i < k; i ++){
    while((int) S.size() > 0 && arr[S.back()] >= arr[i])
      S.pop_back();
    while((int) G.size() > 0 && arr[G.back()] <= arr[i])
      G.pop_back();
    G.push_back(i);
    S.push_back(i);
  }
  for(int i = k; i < n; i ++){
    Sum += arr[S.front()] + arr[G.front()];
    while((int) S.size() > 0 && S.front() <= i - k)
      S.pop_front();
    while((int) G.size() > 0 && G.front() <= i - k)
      G.pop_front();
    while((int) S.size() > 0 && arr[S.back()] >= arr[i])
      S.pop_back();
    while((int) G.size() > 0 && arr[G.back()] <= arr[i])
      G.pop_back();
    G.push_back(i);
    S.push_back(i);
  }
  Sum += arr[S.front()] + arr[G.front()];
  return Sum;
}
```

Figure 2: **Example of unsupervised Python to C++ translation**. TransCoder successfully translates the Python input function `SumOfKsubArray` into C++. TransCoder infers the types of the arguments, of the variables, and the return type of the function. The model maps the Python `deque()` container, to the C++ implementation `deque<>`, and uses the associated `front`, `back`, `pop_back` and `push_back` methods to retrieve and insert elements into the `deque`, instead of the Python square brackets `[]`, `pop` and `append` methods. Moreover, it converts the Python `for` loop and `range` function properly.

successfully pass the unit tests. For instance, when translating from C++ to Java, only 3.1% of the generations are strictly identical to the ground truth reference, although 60.9% of them return the expected outputs. Moreover, the performance in terms of BLEU is relatively flat and does not correlate well with the computational accuracy. These results highlight the issues with the traditional reference match and BLEU metrics commonly used in the field.

**Beam search decoding.** In Table 2, we study the impact of beam search, either by considering all hypotheses in the beam that pass the unit tests (CA@N) or by only considering the ones with the highest log-probabilities (N Beams - CA@1). Compared to greedy decoding (CA@1), beam search significantly improves the computational accuracy, by up to 33.7% in Java → Python with 25 beams (CA@25). When the model only returns the hypothesis with the highest log-probability, the performance drops, indicating that TransCoder often finds a valid translation, although it sometimes gives a higher log-probability to incorrect hypotheses. More generally, beam search allows minor variations of the translations which can make the unit tests succeed, such as changing the return or variable types in Java and C++, or fixing small errors such as the use of / instead of the // operator in Python. More examples of errors corrected by beam search are presented in Figure 9 in the appendix.

In a real use-case, checking whether the generated functions are syntactically correct and compile, or creating unit tests from the input function would be better approaches than comparing log-probabilities in order to select an hypothesis from the beam. Table 5 in the appendix shows that many failures

Table 1: **Results of TransCoder on our test set with greedy decoding.** We evaluate TransCoder with different metrics: reference match, BLEU score, and computational accuracy. Only 3.1% of C++ to Java translations match the ground truth reference, although 60.9% of them successfully pass the unit tests, suggesting that reference match is not an accurate metric to evaluate the quality of translations. Similarly, the BLEU score does not correlate well with the computational accuracy.

| | C++ → Java | C++ → Python | Java → C++ | Java → Python | Python → C++ | Python → Java |
|---|---|---|---|---|---|---|
| Reference Match | 3.1 | 6.7 | 24.7 | 3.7 | 4.9 | 0.8 |
| BLEU | 85.4 | 70.1 | 97.0 | 68.1 | 65.4 | 64.6 |
| Computational Accuracy | 60.9 | 44.5 | 80.9 | 35.0 | 32.2 | 24.7 |

Table 2: **Computational accuracy with beam search decoding and comparison to baselines.** Increasing the beam size improves the performance by up to 33.7% in Java → Python. When the model only returns the hypothesis with the highest log-probability (10 Beams - CA@1), the performance drops, indicating that the model often finds a correct translation, although it does not necessarily assign it with the highest probability. TransCoder significantly outperforms the Java → Python baseline (+30.4%) and the commercial C++ → Java baseline (+13.9%), although it is trained in a fully unsupervised manner and does not leverage human knowledge.

|  | C++ → Java | C++ → Python | Java → C++ | Java → Python | Python → C++ | Python → Java |
|---|---|---|---|---|---|---|
| Baselines | 61.0 | - | - | 38.3 | - | - |
| TransCoder CA@1 | 60.9 | 44.5 | 80.9 | 35.0 | 32.2 | 24.7 |
| TransCoder 10 Beams - CA@1 | 65.1 | 46.9 | 79.8 | 49.0 | 32.4 | 36.6 |
| TransCoder CA@5 | 70.7 | 58.3 | 86.9 | 60.0 | 44.4 | 44.3 |
| TransCoder CA@10 | 73.4 | 62.0 | 89.3 | 64.4 | 49.6 | 51.1 |
| TransCoder CA@25 | 74.8 | 67.2 | 91.6 | 68.7 | 57.3 | 56.1 |

come from compilation errors when the target language is Java or C++. It suggests that the "N Beams - CA@1" metric could easily be improved. We leave this to future work.

**Comparison to existing baselines.** We compare TransCoder with two existing approaches: j2py[10], a framework that translates from Java to Python, and a commercial solution from Tangible Software Solutions[11], that translates from C++ to Java. Both systems rely on rewrite rules manually built using expert knowledge. The latter handles the conversion of many elements, including core types, arrays, some collections (Vectors and Maps), and lambdas. In Table 2, we observe that TransCoder significantly outperforms both baselines in terms of computational accuracy, with 74.8% and 68.7% in the C++ → Java and Java → Python directions, compared to 61% and 38.3% for the baselines. TransCoder particularly shines when translating functions from the standard library. In rule-based transcompilers, rewrite rules need to be manually encoded for each standard library function, while TransCoder learns them in an unsupervised way. In Figure 10 of the appendix, we present several examples where TransCoder succeeds, while the baselines fail to generate correct translations.

### 4.6 Discussion - Analysis

In Figure 2, we give an example of TransCoder unsupervised translation from C++ to Java. Additional examples can be found in Figure 6 and Figure 7 of the appendix. We observe that TransCoder successfully understands the syntax specific to each language, learns data structures and their methods, and correctly aligns libraries across programming languages. For instance, it learns to translate the ternary operator "X ? A : B" in C++ or Java to "if X then A else B" in Python, in an unsupervised way. In Figure 5 of the appendix, we present a t-SNE [35] visualization of cross-lingual token embeddings learned by the model. TransCoder successfully map tokens with similar meaning to the same latent representation, regardless of their languages. Figure 8 of the appendix shows that TransCoder can adapt to small modifications. For instance, renaming a variable in the input may result in different translated types, still with valid translations. In Figure 11, we present some typical failure cases where TransCoder fails to account for the variable type during generation. For instance, it copies the C++ NOT operator `!` applied to an integer in Java, while it should be translated to `~`. It also translates the Python `min` function on lists to `Math.min` in Java, which is incorrect when applied to Java arrays. Finally, Table 5 gives detailed results on failure cases, and Table 7 gives the model accuracy for different function lengths.

## 5 Conclusion

In this paper, we show that approaches of unsupervised machine translation can be applied to source code to create a transcompiler in a fully unsupervised way. TransCoder can easily be generalized to any programming language, does not require any expert knowledge, and outperforms commercial solutions by a large margin. Our results suggest that a lot of mistakes made by the model could easily be fixed by adding simple constraints to the decoder to ensure that the generated functions are syntactically correct, or by using dedicated architectures [12]. Leveraging the compiler output or other approaches such as iterative error correction [16] could also boost the performance.

## Broader Impact

Automatic transcompilation has the potential to make programmers working in companies or on open source projects more efficient, by allowing them to integrate various codes from other teams within the company or other open source projects more easily. It can also lower the cost of updating an old codebase written in an obsolete language to a more recent language. Many large banks, insurance companies and utilities companies still run code written in COBOL. Advances in transcompilation could incite them to update to more recent languages and facilitate future innovation. Transcompilation being a tool facilitating innovation, its applications could have both positive and negative societal impacts. However, we believe that the impact of more innovation within companies and in open source projects would be positive overall. In the long-term, updates of old codebases could put the experts in obsolete languages at a disadvantage as the need for their expertise will decrease. We believe it would be offset by an increased need for programmers caused by more innovative projects, benefiting all programmers.

## Funding Disclosure

The authors of this paper are employed by Facebook France and this work was done using hardware and software provided by Facebook.

## Footnotes

[2]https://docs.python.org/2/library/2to3.html

[3] In practice, the "cross-linguality" of the model highly depends on the amount of anchor points across languages. As a result, a XLM model trained on English-French will provide better cross-lingual representations than a model trained on English-Chinese, because of the different alphabet which reduces the number of anchor points. In programming languages, the majority of strings are composed of English words, which results in a fairly high number of anchor points, and the model *naturally* becomes cross-lingual.

[4]https://console.cloud.google.com/marketplace/details/github/github-repos

[5]`https://github.com/c2nes/javalang`

[6]`https://docs.python.org/3/library/tokenize.html`

[7]`https://pypi.org/project/clang`

[8]`https://github.com/glample/fastBPE`

[9]`https://practice.geeksforgeeks.org`

[10]`https://github.com/natural/java2python`

[11]`https://www.tangiblesoftwaresolutions.com/`

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
