[Supplementary Material]

# A  Data and preprocessing

## A.1  Training dataset

We tried removing and keeping the comments in the code from our training data. As shown in Table 6, keeping the comments gives better results overall. Thus, we decided to keep them in our final training data. Detailed statistics of the resulting dataset can be found in Table 3.

Table 3: **Statistics of our GitHub dataset.** We show the statistic for our entire github dataset (All) and for the extracted functions. We give the size in GigaBytes, the number of files and functions, and the number of tokens.

|  | C++ | Java | Python |
|---|---|---|---|
| All - Size | 168 GB | 352 GB | 224 GB |
| All - Nb of files | 15 M | 56 M | 18 M |
| All - Nb of tokens | 38 B | 75 B | 50 B |
| Functions - Size | 93 GB | 185 GB | 152 GB |
| Functions - Nb of functions | 120 M | 402 M | 217 M |

## A.2  Tokenization

| Python function v1 | Python function v2 |
|---|---|

```python
def rm_file(path):
    try:
        os.remove(path)
        print("Deleted")
    except:
        print("Error while deleting file", path)
```

```python
def rm_file(path):

    try:
        os.remove( path )
        print( "Deleted" )
    except  :
        print("Error while deleting file", path)
```

```
def rm_file ( path ) : NEWLINE try : NEWLINE INDENT os . remove (path) NEWLINE print ( " Deleted " )
DEDENT except : NEWLINE INDENT print ( " Error _ while _ deleting _ file " , path ) DEDENT
```

Figure 3: **Example of function tokenization.** We show two versions of the same Python function and their common tokenization. These function versions differ by extra spaces and one extra new line. Our Python tokenizer is robust to extra spaces and extra new lines except in strings. In strings, spaces are tokenized as ▁ (U+2581). Indentation is meaningful in Python: indented blocks are surrounded by INDENT DEDENT tokens.

## A.3  Function extraction

We train and evaluate our translation model on functions only. We differentiate class functions and standalone functions. By standalone functions, we refer to functions that can be used without instantiating a class. In C++ and Python, this corresponds to static methods of classes, and functions outside classes. In Java, it only corresponds to static methods. In GeeksforGeeks, solutions are implemented with standalone functions, and our evaluation protocol only involves these functions. In Table 3, the functions statistics are given for all kind of functions. In C++ and Python, 50% of functions are standalone functions. In Java, standalone functions only represent 15% of the dataset. We tried to train our model on standalone functions only, and observed better results than when training on all functions. Thus, all the results in this work are given for models pretrained on all available data and trained on standalone functions only.

# B  Evaluation

GeeksforGeeks is an online platform with computer science and programming articles. It contains many coding problems and presents solutions in several programming languages. We gather all the problems for which some solutions are implemented in C++, Java, and Python. The parallel data for these problems is already enough to test a model using the BLEU score or the Reference Match score. However, we need to generate some unit tests to check that the function are semantically correct and to compute the Computational Accuracy.

These unit tests are contained in a script, which contains a reference function — named `f_gold` — from the parallel dataset, a commented TOFILL marker which is to be replaced with a generated function, and a main which runs both functions on a series of inputs and compares the behaviors of the two functions. We have one script per function and per programming language.

In order to generate these scripts, we extract the parameters and their types from the Java implementation of the solution. Then, we generate 10 random inputs for these types, which are hardcoded in the test script and used to test the function. We test the generated scripts by injecting the reference function a second time with the name `f_filled` instead of the TOFILL comment and running it. We keep only the scripts that return a perfect score in less than 10 seconds. As Python is dynamically typed, we need to infer the Python parameters types from the Java types, and to assume that the order and types of the parameters is the same in Java and Python. When this assumption happens to be wrong, the generated script fails the tests and is discarded. As this approach is quite effective, we generated the C++ scripts in a similar manner and barely use the C++ parameter types which can be extracted from the function definition.

**Equality tests.**   We adapt the tests checking that the reference and gold function behave in the same way based on the output type of the function (extracted from its Java implementation). For instance, we test the equality of `int` outputs with `==`, while we use `equals` for `String` outputs and relative tests for `double` outputs. If the function is inplace (the output type is `void`), we check the side effects on all its mutable arguments instead.

**Special cases for random input generation.**   The goal of our scripts is to decide whether a function is semantically equivalent to from the reference function, and the way we generate the random inputs is critical to how discriminative the script will be. For instance, if the input of the reference function is a string, a naive solution may be to generate strings of random length and with characters sampled randomly from the set of all characters. However, our dataset contains several functions such as `checkDivisibility` in Figure 4 which considers the string to be a representation of a long integer. This type of function could always return the same result (e.g. `False`) on inputs strings that do not contain only digits. As many functions in our dataset assume the input strings or characters to be representations of long integers or representations of integers in base 2, we alternate between sampling the characters from (i) the set of all lowercase and uppercase letters plus the space character, (ii) the set of all digits, and (iii) the set containing 0 and 1. For similar reasons, when there is an integer array in the function parameters, we alternate between the sets $\{0\ldots100\}$, $\{-100\ldots100\}$ and $\{0, 1\}$ to sample the integers inside the array. When the function takes no argument, we do not generate any input for it and only check that the output is the same for the reference function and the generated function.

**Manual verifications.**   In order to ensure that our unit tests are appropriate, we manually check and modify the scripts when the output of the function is the same on all 10 inputs, when the function is inplace, or when the function contains prints. As we only check the side effects affecting the mutable arguments, we remove all the functions which mainly print or write to a file.

| C++ | Java | Python |
|---|---|---|

```cpp
bool checkDivisibility(string num){
  int length = num.size();
  if(length == 1 && num[0] == '0')
    return true;
  if(length % 3 == 1){
    num += "00";
    length += 2;
  }
  else if(length % 3 == 2){
    num += '0';
    length += 1;
  }

  int sum = 0, p = 1;
  for(int i = length - 1;
          i >= 0; i--){
    int group = 0;
    group += num[i--] - '0';
    group += (num[i--] - '0') * 10;
    group += (num[i] - '0') * 100;
    sum = sum + group * p;
    p *= (-1);
  }

  sum = abs(sum);
  return (sum % 13 == 0);
}
```

```java
static boolean checkDivisibility(
                String num){
  int length = num.length();
  if(length == 1 && num.charAt(0) == '0')
    return true;
  if(length % 3 == 1){
    num += "00";
    length += 2;
  }
  else if(length % 3 == 2){
    num += "0";
    length += 1;
  }

  int sum = 0, p = 1;
  for(int i = length - 1; i >= 0; i--){
    int group = 0;
    group += num.charAt(i--) - '0';
    group += (num.charAt(i--) - '0') * 10;
    group += (num.charAt(i) - '0') * 100;
    sum = sum + group * p;
    p *= (-1);
  }

  sum = Math.abs(sum);
  return (sum % 13 == 0);
}
```

```python
def checkDivisibility(num):
  length = len(num)
  if(length == 1 and num[0] == '0'):
    return True
  if(length % 3 == 1):
    num = str(num) + "00"
    length += 2
  elif(length % 3 == 2):
    num = str(num) + "0"
    length += 1

  sum = 0
  p = 1
  for i in range(length - 1, -1, -1):
    group = 0
    group += ord(num[i]) - ord('0')
    i -= 1
    group += (ord(num[i]) - ord('0')) * 10
    i -= 1
    group += (ord(num[i]) - ord('0')) * 100
    sum = sum + group * p
    p *= (-1)

  sum = abs(sum)
  return (sum % 13 == 0)
```

Figure 4: **Example of parallel function from our test set.** We extracted parallel functions from GeeksforGeeks to create validation and test sets. Here, we have the parallel implementations in C++, Java, and Python of the checkDivisibility function, which determines whether a long integer represented as a string is divisible by 13.

Table 4: **Number of functions with unit tests for our validation and test sets.** We report the number of function with unit tests for C++, Java, and Python, for the validation and test sets. We also show the average number of tokens per function. A unit test checks whether a generated function is semantically equivalent to its reference. For each function, we have 10 unit tests, each testing it on a different input. As a result, the number of functions with unit tests per language gives the size of the validation and test sets of each pair of languages. For instance, we have 231 C++ functions with unit tests for the validation set, which means that we have a validation set of 231 functions for Java → C++ and Python → C++.

|  | C++ | Java | Python |
|---|---|---|---|
| Nb of functions with unit tests - valid set | 231 | 234 | 237 |
| Nb of functions with unit tests - test set | 466 | 481 | 463 |
| Average #tokens per function | 105.8 | 112.0 | 103.1 |

# C Results

## C.1 Detailed results

Table 5: **Detailed results for greedy decoding.** Many failures come from compilation errors when the target language is Java or C++. It suggests that our method could be improved by constraining the decoder to generate compilable code. Runtime errors mainly occur when translating from Java or C++ into Python. Since Python code is interpreted and not compiled, this category also includes syntax errors in Python. The majority of remaining errors are due to the program returning the wrong output on one or several of the unit tests. Timeout errors are generally caused by infinite loops and mainly occur in the Java $\leftrightarrow$ Python pair.

| | #tests | Success | Compilation | Runtime | Wrong Output | Timeout |
|---|---|---|---|---|---|---|
| C++ $\rightarrow$ Java | 481 | 60.9% | 27.2% | 4.4% | 5.4% | 2.1% |
| C++ $\rightarrow$ Python | 463 | 44.5% | 0.0% | 36.5% | 18.1% | 0.9% |
| Java $\rightarrow$ C++ | 466 | 80.9% | 10.3% | 1.1% | 7.5% | 0.2% |
| Java $\rightarrow$ Python | 463 | 35.0% | 0.0% | 31.8% | 15.6% | 17.7% |
| Python $\rightarrow$ C++ | 466 | 32.2% | 29.0% | 4.9% | 32.6% | 1.3% |
| Python $\rightarrow$ Java | 481 | 24.7% | 23.5% | 12.5% | 24.3% | 15.0% |

## C.2 Ablation study

Table 6: **Training data ablation study - with and without code comments.** We compare the computational accuracy of TransCoder for different training sets, where we either keep or remove comments from source code training data. We give results for different beam sizes. When translating from C++ to Python, from Java to C++ and from Java to Python, keeping comments in the training set gives better results. In the other directions, keeping or removing comments does not have a significant impact on the performance.

| With Comments | C++ $\rightarrow$ Java | | C++ $\rightarrow$ Python | | Java $\rightarrow$ C++ | | Java $\rightarrow$ Python | | Python $\rightarrow$ C++ | | Python $\rightarrow$ Java | |
|---|---|---|---|---|---|---|---|---|---|---|---|---|
| | No | Yes | No | Yes | No | Yes | No | Yes | No | Yes | No | Yes |
| CA@1 | 62.2 | 60.9 | 40.8 | 44.5 | 76.8 | 80.9 | 46.4 | 35.0 | 34.1 | 32.2 | 33.9 | 24.7 |
| CA@5 | 71.6 | 70.7 | 54.0 | 58.3 | 85.6 | 86.9 | 58.5 | 60.0 | 46.4 | 44.4 | 46.0 | 44.3 |
| CA@10 | 73.6 | 73.4 | 57.9 | 62.0 | 88.4 | 89.3 | 62.9 | 64.4 | 50.9 | 49.6 | 50.3 | 51.1 |
| CA@25 | 75.3 | 74.8 | 64.6 | 67.2 | 89.1 | 91.6 | 66.7 | 68.7 | 56.7 | 57.3 | 56.3 | 56.1 |

## C.3 Cross-lingual token embedding space

Figure 5: **Cross-lingual token embedding space.** We show a t-SNE visualization of our cross-lingual token embeddings. These embeddings are obtained by encoding programming language tokens into TransCoder's lookup table. We show the embeddings of C++, Java, and Python keywords. Keywords of different programming languages that are used in similar contexts are very close in the embedding space. For instance, `except` in Python and `catch` in Java and C++, which are both used to catch exceptions, are mapped to very similar embeddings. The same phenomenon is observed for implementations of maps (`Map`, `map` and `dict`), for `c_str` and `toCharArray` which are used to transform a string into a char array, and for similar primitive types (e.g. `Long`, `long`, `Integer`, and `int`).

## C.4 Additional examples of translations

| C++ Input | Unsupervised Java Translation |
|---|---|
| <pre>int maxLen(int arr[], int n){<br>  unordered_map<int, int> hM;<br>  int sum = 0;<br>  int max_len = 0;<br>  int ending_index = - 1;<br>  for(int i = 0; i < n; i++)<br>    arr[i] = (arr[i] == 0) ? -1 : 1;<br>  for(int i = 0; i < n; i++){<br>    sum += arr[i];<br>    if(sum == 0){<br>      max_len = i + 1;<br>      ending_index = i;<br>    }<br>    if(hM.find(sum + n) != hM.end()){<br>      if(max_len < i - hM[sum + n]){<br>        max_len = i - hM[sum + n];<br>        ending_index = i;<br>      }<br>    }<br>    else hM[sum + n] = i;<br>  }<br>  for(int i = 0; i < n; i++)<br>    arr[i] = (arr[i] == -1) ? 0 : 1;<br>  printf("%d to %d\n",<br>         ending_index - max_len + 1,<br>         ending_index);<br>  return max_len;<br>}</pre> | <pre>public static int maxLen(int arr[], int n){<br>  HashMap<Integer, Integer> hM<br>         = new HashMap<Integer, Integer>();<br>  int sum = 0;<br>  int maxLen = 0;<br>  int endingIndex = - 1;<br>  for(int i = 0; i < n; i++){<br>    arr[i] = (arr[i] == 0) ? - 1 : 1;<br>  }<br>  for(int i = 0; i < n; i++){<br>    sum += arr[i];<br>    if(sum == 0){<br>      maxLen = i + 1;<br>      endingIndex = i;<br>    }<br>    if(hM.containsKey(sum + n)){<br>      if(maxLen < i - hM.get(sum + n)){<br>        maxLen = i - hM.get(sum + n);<br>        endingIndex = i;<br>      }<br>    }else{<br>      hM.put(sum + n, i);<br>    }<br>  }<br>  for(int i = 0; i < n; i ++){<br>    arr[i] = (arr[i] == -1) ? 0 : 1;<br>  }<br>  System.out.println(endingIndex - maxLen + 1<br>                     + " to " + endingIndex);<br>  return maxLen;<br>}</pre> |
| <pre>std::unordered_set<int> uniqueElements(<br>                const std::vector<int>& l){<br>  return std::unordered_set<int>(l.begin(), l.end());<br> }</pre> | <pre>public static Set<Integer> uniqueElements(<br>                Collection<Integer> l){<br>  return new HashSet<Integer>(l);<br> }</pre> |
| <pre>std::vector<int> squareList(const std::vector<int>& l){<br>  std::vector<int> result = {};<br>  for(int elmt : l){<br>    result.push_back(elmt * elmt);<br>  }<br>  return result;<br> }</pre> | <pre>public static List<Integer> squareList(<br>                List<Integer> l){<br>  List<Integer> result = new ArrayList<>();<br>  for(Integer elmt : l) result.add(elmt*elmt);<br>  return result;<br>}</pre> |

Figure 6: **Examples of correct translations using TransCoder**. In all these examples, TransCoder converts properly the primitive types, find the equivalent data structures and the corresponding methods, and libraries between languages. For instance, in the maxLen function, TransCoder converts the C++ unordered_map container into the Java implementation of the Map interface, HashMap, and properly uses the associated get and put methods to retrieve and insert keys and values into the map, instead of the C++ square brackets [].

| Input | TransCoder Translation |
|---|---|

### C++ → Python

```cpp
void worstFit(int blockSize[], int m, int processSize[], int n){
  int allocation[n];
  memset(allocation, -1, sizeof(allocation));
  for(int i = 0; i < n; i ++){
    int wstIdx = - 1;
    for(int j = 0; j < m; j++){
      if(blockSize[j] >= processSize[i]){
        if(wstIdx == - 1)
          wstIdx = j;
        else if(blockSize[wstIdx] < blockSize[j])
          wstIdx = j;
      }
    }
    if(wstIdx != -1){
      allocation[i] = wstIdx;
      blockSize[wstIdx] -= processSize[i];
    }
  }
  cout << "\nProcess No.\tProcess Size\tBlock no.\n";
  for(int i = 0; i < n; i++){
    cout << "   " << i+1 << "\t\t" << processSize[i] << "\t\t";
    if(allocation[i] != -1) cout << allocation[i] + 1;
    else
      cout << "Not Allocated";
  cout << endl;
  }
}
```

```python
def worstFit(blockSize, m, processSize, n):
  allocation = [-1] * n
  for i in range(n):
    wstIdx = -1
    for j in range(m):
      if blockSize[j] >= processSize[i]:
        if wstIdx == -1:
          wstIdx = j
        elif blockSize[wstIdx] < blockSize[j]:
          wstIdx = j
    if wstIdx != -1:
      allocation[i] = wstIdx
      blockSize[wstIdx] -= processSize[i]
  print("\nProcess No.\tProcess Size\tBlock no.\n")
  for i in range(n):
    print("   " + str(i+1) + "\t\t" +
        str(processSize[i]) + "\t\t")
    if allocation[i] != -1:
      print(allocation[i] + 1)
    else:
      print("Not Allocated")
  print()
```

### Java → Python

```java
public static int max(int a, int b){
  return a > b ? a : b;
}
```

```python
def max(a, b):
  return a if a > b else b
```

```java
public static void createDirectory(Path path)
        throws IOException{
  if(!Files.exists(path)){
    Files.createDirectories(path);
  }
}
```

```python
def create_directory(path):
  if not os.path.exists(path):
    os.makedirs(path)
```

### Python → C++

```python
def sum_elements(values):
    return sum(values)
```

```cpp
template <typename T> T
sum_elements(const std::vector<T> &values){
  return std::accumulate(values.begin(), values.end(),
                            0);
}
```

```python
def no_letters(s):
  return s.lower() == s.upper()
```

```cpp
static bool noLetters(const QString& str){
  return str.toLower() == str.toUpper();
}
```

```python
def get_env_variable(name):
  return os.environ[name]
```

```cpp
const char* GetEnvVariable(const char* name){
  return getenv(name);
}
```

### Python → Java

```python
def calcMaxValue(str):
  res = ord(str[0]) - 48
  for i in range(1, len(str)):
    if(str[i] == '0' or str[i] == '1' or res < 2):
      res += ord(str[i]) - 48
    else:
      res *= ord(str[i]) - 48
  return res
```

```java
public static int calcMaxValue(String str){
  int res = (int)(str.charAt(0) - 48);
  for(int i = 1; i < str.length(); i++){
    if(str.charAt(i) == '0'
        || str.charAt(i) == '1'
        || res < 2){
      res += (int)(str.charAt(i) - 48);
    }
    else{
      res *= (int)(str.charAt(i) - 48);
    }
  }
  return res;
}
```

```python
def foo(x):
  return bar(x) + 1
```

```java
public static int foo(int x){
  return bar(x) + 1;
}
```

```python
def area(r):
  return 2 * PI * r ** 2
```

```java
public static double area(double r){
  return 2 * PI * r * r;
}
```

Figure 7: **Examples of correct translations from or to Python using TransCoder**. When translating from Python, TransCoder successfully infers types. Here, TransCoder infers the Python `list` type and translates it into its C++ equivalent `std::vector`. The last two examples show that TransCoder does not modify the call to the non-standard function `bar` or the global variable `PI`.

| C++ input | TransCoder Java translation |
|---|---|

```cpp
int minPalPartion(char* str){
  int n = strlen(str);
  int C[n][n];
  bool P[n][n];
  int i, j, k, L;
  for(i = 0; i < n; i ++){
    P[i][i] = true;
    C[i][i] = 0;}
  for(L = 2; L <= n; L ++){
    for(i = 0; i < n - L + 1; i ++){
      j = i + L - 1;
      if(L == 2) P[i][j] = (str[i] == str[j]);
      else P[i][j] = (str[i] == str[j])
              && P[i+1][j-1];
      if(P[i][j] == true) C[i][j] = 0;
      else{
        C[i][j] = INT_MAX;
        for(k = i; k <= j - 1; k ++)
            C[i][j] = min(C[i][j],
                    C[i][k]+C[k+1][j]+1);
      }
    }
  }
  return C[0][n - 1];
}
```

```java
public static int minPalPartion(String str){
  int n = str.length();
  int[][] C = new int[n][n];
  boolean[][] P = new boolean[n][n];
  int i , j , k , L;
  for(i = 0; i < n; i ++){
    P[i][i] = true;
    C[i][i] = 0;}
  for(L = 2; L <= n; L ++){
    for(i = 0; i < n - L + 1;  i ++){
      j = i + L - 1;
      if(L == 2){
        P[i][j] = (str.charAt(i) == str.charAt(j));}
      else{
        P[i][j] = (str.charAt(i) == str.charAt(j))
                && P[i + 1][j - 1];}
      if(P[i][j] == true){
        C[i][j] = 0;}
      else{
        C[i][j] = Integer.MAX_VALUE;
        for(k = i; k <= j - 1; k ++){
          C[i][j] = Math.min(C[i][j],
                  C[i][k] + C[k + 1][j] + 1);}}}}
  return C[0][n - 1];
}
```

```cpp
int minPalPartion(char* arr){
  int n = strlen(arr);
  int C[n][n];
  bool P[n][n];
  int i, j, k, L;
  for(i = 0; i < n; i ++){
    P[i][i] = true;
    C[i][i] = 0;}
  for(L = 2; L <= n; L ++){
    for(i = 0; i < n - L + 1; i ++){
      j = i + L - 1;
      if(L == 2) P[i][j] = (arr[i] == arr[j]);
      else P[i][j]= (arr[i] == arr[j])
              && P[i+1][j-1];
      if(P[i][j] == true) C[i][j] = 0;
      else{
        C[i][j] = INT_MAX;
        for(k = i; k <= j - 1; k ++)
            C[i][j] = min(C[i][j],
                    C[i][k]+C[k+1][j]+1);
      }
    }
  }
  return C[0][n - 1];
}
```

```java
public static int minPalPartion(char[] arr){
  int n = arr.length;
  int[][] C = new int[n][n];
  boolean[][] P = new boolean[n][n];
  int i , j , k , L;
  for(i = 0; i < n; i ++){
    P[i][i] = true;
    C[i][i] = 0;}
  for(L = 2; L <= n; L ++){
    for(i = 0; i < n - L + 1;  i ++){
      j = i + L - 1;
      if(L == 2){
        P[i][j] = (arr[i] == arr[j]);}
      else{
        P[i][j] = (arr[i] == arr[j])
                && P[i + 1][j - 1];}
      if(P[i][j] == true){
        C[i][j] = 0;}
      else{
        C[i][j] = Integer.MAX_VALUE;
        for(k = i; k <= j - 1; k ++){
          C[i][j] = Math.min(C[i][j],
                  C[i][k] + C[k + 1][j] + 1);}}}}
  return C[0][n - 1];
}
```

```cpp
int minPalPartion(char* input){
  int n = strlen(input);
  int C[n][n];
  bool P[n][n];
  int i, j, k, L;
  for(i = 0; i < n; i ++){
    P[i][i] = true;
    C[i][i] = 0;}
  for(L = 2; L <= n; L ++){
    for(i = 0; i < n - L + 1; i ++){
      j = i + L - 1;
      if(L == 2) P[i][j] = (input[i] == input[j]);
      else P[i][j] = (input[i] == input[j])
              && P[i+1][j-1];
      if(P[i][j] == true) C[i][j] = 0;
      else{
        C[i][j] = INT_MAX;
        for(k = i; k <= j - 1; k ++)
            C[i][j] = min(C[i][j],
                    C[i][k]+C[k+1][j]+1);
      }
    }
  }
  return C[0][n - 1];
}
```

```java
public static int minPalPartion(String input){
  int n = input.length();
  int[][] C = new int[n][n];
  boolean[][] P = new boolean[n][n];
  int i , j , k , L;
  for(i = 0; i < n; i ++){
    P[i][i] = true;
    C[i][i] = 0;}
  for(L = 2; L <= n; L ++){
    for(i = 0; i < n - L + 1;  i ++){
      j = i + L - 1;
      if(L == 2){
        P[i][j] = (input.charAt(i) == input.charAt(j));
      }
      else{
        P[i][j] = (input.charAt(i) == input.charAt(j))
                && P[i + 1][j - 1];}
      if(P[i][j] == true){
        C[i][j] = 0;}
      else{
        C[i][j] = Integer.MAX_VALUE;
        for(k = i; k <= j - 1; k ++){
          C[i][j] = Math.min(C[i][j],
                  C[i][k] + C[k + 1][j] + 1);}}}}
  return C[0][n - 1];
}
```

Figure 8: **TransCoder robustness to variable names.** We take the C++ function `minPalPartion`, change the parameter name from `str` to `arr` and `input` and show the three corresponding TransCoder Java translations. All translations are correct. In the first and third cases, TransCoder translates `char* str` and `char* input` into Java `String` and uses the `charAt` method to retrieve elements. This shows that TransCoder is robust to variable name changes and that it remembers variable types along the whole translation to apply the appropriate methods. In the second case, TransCoder translates `char* arr` into Java `char[]` and uses `[]` to retrieve elements, showing that TransCoder can adjust its translations to parameter names while remaining accurate.

| C++ input | TransCoder Python translation | |
|---|---|---|
| | Greedy decoding | Beam search |

```cpp
int compute_average(int a, int b){
  return (a + b) / 2;
}
```

```python
def compute_average(a, b):
  return (a + b) / 2
```

```python
def compute_average(a, b):
  return (a + b) // 2
```

```cpp
bool isPalindrome(string str){
  int l = 0;
  int h = str.length() - 1;
  while(h > l)
    if(str[l++] != str[h--])
      return false;
  return true;
}
```

```python
def isPalindrome(str):
  l = 0
  h = len(str) - 1
  while h > l:
    if str[l] != str[h--]:
      return False
  return True
```

```python
def isPalindrome(str):
  l = 0
  h = len(str) - 1
  while h > l:
    if str[l] != str[h]:
      return False
    l += 1
    h -= 1
  return True
```

Figure 9: **Examples of incorrect greedy decoding translations versus correct beam translations.** We take C++ inputs and translate them into Python with TransCoder. In the second column, we use greedy decoding and the translations are incorrect. In the third column, we use beam search and obtain accurate translations. A common error corrected by beam search for C++ $\rightarrow$ Python is the usage of the double instead of the integer division operator (first example). Beam search is also able to correct errors such as the $++$ and $--$ operators that do not exist in Python (second example).

| Input | Baseline translation | TransCoder translation |
|---|---|---|
| Java | Python | Python |

```java
static String getEnvVariable(
        String name){
  return System.getenv(name);
}
```

```python
def getEnvVariable(name):
    return System.getenv(name)
```

```python
def getenvh(name):
    return os.environ.get(name, '')
```

```java
static List<Integer> truncate(
    List<Integer> l,
    Integer e){
  if(l.contains(e)){
    return l.subList(0,
        l.indexOf(e));
  }
  else{
    return l;
  }
}
```

```python
def truncate(l, e):
    if l.contains(e):
        return l.subList(0,
                l.indexOf(e))
    else:
        return l
```

```python
def truncate(l, e):
    if l.count(e) > 0:
        return l[ : l.index(e)]
    else:
        return l
```

```java
static void deleteFile(Path path){
  try{
    Files.delete(path);
  }
  catch(Exception e){
    System.err.println(
    "Error deleting " + path);
  }
}
```

```python
def deleteFile(path):
    try:
        Files.delete(path)
    except Exception as e:
        System.err.println(
        "Error deleting " + path)
```

```python
def delete_file(path):
    try:
        os.remove(path)
    except:
        sys.stderr.write(
            "Error deleting %s\n"
            % path
        )
```

| C++ | Java | Java |
|---|---|---|

```cpp
memset(prime, 0, sizeof(prime));
```

```java
memset(prime, 0,
    (Integer.SIZE/Byte.SIZE));
```

```java
Arrays.fill(prime, 0);
```

```cpp
sort(a, a + n);
```

```java
sort(a, a + n);
```

```java
Arrays.sort(a);
```

```cpp
for(char ch : str)
```

```java
for(char ch : str)
```

```java
for(char ch : str.toCharArray())
```

Figure 10: **Examples of incorrect baseline translations versus correct TransCoder translations.** When translating from Java to Python, the baseline fails to translate the `System.getenv`, `System.err.println`, and `Files.delete` functions from the standard library, and the `contains`, `subList`, and `IndexOf` methods of the Java `List` interface. Instead, it simply copies them, showing the limitations of a rule-based system. On the other hand, TransCoder converts properly all of these functions into their Python equivalents. In the C++ $\rightarrow$ Java direction, baseline translations are made at token-level, and are incorrect. For instance, the first example shows that the baseline tries to translate the `sizeof` function, and leaves `memset` unchanged although it does not exist in Java. Instead, TransCoder correctly uses `Arrays.fill` to fill the array `prime` with zeros.

Table 7: **Performance v.s. function lengths.** CA@1 metric for various function lengths (in number of tokens) for C++ → Python with 10 beams.

| Number of Tokens | CA@1 |
|---|---|
| $0 \leq n < 30$ | 93.0% |
| $30 \leq n < 50$ | 91.7% |
| $50 \leq n < 61$ | 93.5% |
| $61 \leq n < 72$ | 71.7% |
| $72 \leq n < 85$ | 70.2% |
| $85 \leq n < 99$ | 56.3% |
| $99 \leq n < 121$ | 37.0% |
| $121 \leq n < 147$ | 42.2% |
| $147 \leq n < 179$ | 39.6% |
| $179 \leq n < 326$ | 28.3% |

| Input | Java failed translations | Description |
|---|---|---|
| ```bool isEven (int n){```<br>```  return (!(n & 1));```<br>```}``` | ```static boolean isEven(int n){```<br>```  return (!(n & 1));```<br>```}``` | The `!` operator works on boolean and integers in C++ (it returns `true` if the integer is positive) but it only works on boolean in Java. |
| ```int summingSeries(long n){```<br>```  return pow(n, 2);```<br>```}``` | ```static int summingSeries(long n){```<br>```  return Math.pow(n, 2);```<br>```}``` | In Java, `Math.pow(n, 2)` returns a `double` which should be cast to `int` to match the function return type. |
| ```def minSum(A):```<br>```    min_val = min(A)```<br>```    return min_val * (len(A) - 1)``` | ```static double minSum(double[] A){```<br>```  double minVal = Math.min(A);```<br>```  return minVal*(A.length - 1);```<br>```}``` | `Math.min` is a Java function but does not take as input a `double[]` array but a pair of `double`. |

Figure 11: **Examples of failed TransCoder translations.** TransCoder fails to translate these C++ and Python functions into Java, showing its limitations. In these examples, it fails to account for the variable types when using a method or an operator. In particular, the NOT operator `!` in C++ should have been translated to `~` in Java, because it is applied to an integer. Similarly, the `Math.min` function in Java cannot be applied to arrays.