[Reviews · NeurIPS 2020]

Review 1

Summary and Contributions: This paper applies the unsupervised machine translation method of Lample et al (and Artetxe et al) to the task of translating from one programming language to another.

Strengths: The main contribution of this work is just showing that this can be done, which is good to see. The results are good compared to a rule-based baseline. The other important contribution this work makes is to show that BLEU is not an appropriate evaluation metric, and a more extrinsic measure (% of test cases passed) is needed.

Weaknesses: Technically the method only makes small changes from the method of Lample et al., e.g., in the handling of tokenization.

Correctness: I would like to take issue with Table 2, which compares "Baselines" against what appear to be several comparable methods, like "Transcoder Beam 1" and "TransCoder Beam 5". The improvements that come from varying the beam size are huge (60.9 -> 70.7 going from "Beam 1" to "Beam 5"), which confused me initially. Two distinctions should be made more clearly. First, beam decoding and top-k decoding are not the same thing, although in the NMT world they are nearly the same thing and are often confused. In decoders that use dynamic programming (like the Viterbi algorithm for HMMs), the hypotheses in the beam at the last time step are not the top k hypotheses. Making this distinction more clearly would shed light on the second distinction, which is that varying the beam size is actually a variation in the evaluation metric, not the method, so that "Beam 1" scores are not directly comparable to "Beam 5" scores. Can I suggest instead calling these "CA@1", "CA@5", etc., to underscore that these are really different evaluation metrics? OTOH, "Beam 1" vs "Beam 10 - Top 1" is a direct comparison. I would say in this case that the methods being compared are "Beam 1" vs "Beam 10" and the metric is "CA@1". ETA: Thanks for your response; I'm glad you agree with these suggestions.

Clarity: Yes, except please see "Correctness" above.

Relation to Prior Work: The dependence of this work on Lample et al. is made very clear. The correct citation for Artetxe et al "Unsupervised statistical machine translation" is Proceedings of EMNLP 2018.

Reproducibility: Yes

Additional Feedback: The Broader Impact section focuses only on positives; aren't there potential negative consequences of relying on ML-generated code?


Review 2

Summary and Contributions: The paper proposes an unsupervised method for computer program to program transliteration using approaches from unsupervised machine translation literature. More specifically, the paper investigates transliteration of functions in programming languages such as C++, Java, or Python to each other.

Strengths: - The paper is able to obtain strong results in the challenging task of program transliteration using fully unsupervised learning approaches, which is the biggest strength of the paper. - Although these unsupervised methods of masked language model pre-training, denoising autoencoding, and backtranslation are already well-established and have are commonly used on mainstream NLP tasks, their successful application to program transliteration is first shown by this paper. - I feel that due to the strong results obtained the paper, the community will find the methods and results of the paper very useful in future work in the program transliteration field. This is a very nice piece of work. Based on the above strengths of the work, I recommend its acceptance.

Weaknesses: However, the paper has some weakness as well, which are detailed here: - There are no detailed results/comparisons with supervised approaches as baselines in the Results section. These comparisons would have helped to understand the limits of the unsupervised methods. One reason might be the lack of high-quality parallel corpora to train the models with, but nevertheless such results would be quite valuable in further improving the understanding of the limitations/strengths of unsupervised approaches. - The paper evaluates to translate functions with an average length of 110 tokens. It would have been good to have the results on longer functions, or a collections of functions in a file. - The paper doesn't provide ablation studies of the relative importance of different pre-training steps such as the importance of the masked LM using XLM pre-training step? Can this be substituted by the pretraining strategy of models like T5 or BART. In the case of T5 or BART, can the first two steps (masked LM and denoising auto-encoding) be combined into a common step?

Correctness: Yes, I find that the claims, methods, and empirical methodology to be sound and correct.

Clarity: Yes, the paper is very clear, easily understandable and well-written. Kudos to the authors!

Relation to Prior Work: Yes, the paper discusses prior work extensively both with respect to the task and approach-level.

Reproducibility: Yes

Additional Feedback: [UPDATE] In the rebuttal, the authors plan to revise the paper to include more experiments for ablation studies and comparisons with supervised baselines. Overall, I am positive about the paper and the usefulness of its results to the research community.


Review 3

Summary and Contributions: The paper presents a technique for unsupervised translation between programming languages. The paper borrows heavily from Lample et al. [32] where cross-lingual language models have been trained for translation between natural languages. It follows the same general flow for using large monolingual corpora of each language to obtain the NMT model. The paper presents TransCoder, a system for translation between programming languages based on monolingual source code. The system is evaluated over language pairs from (C++, Java, Python). TransCoder outperforms commercial rules-based baselines.

Strengths: - a simple approach, lifting results from NLP XLM to programming languages - outperform commercial rule-based translation systems - results are impressive - a thorough analysis of the results and sources of errors

Weaknesses: This is a nice experiment but I had hoped to learn more about what are the unique challenges for translation of programming languages and what had to be done to addresses these challenges. I guess there is just as much that you can fit into a single paper.

Correctness: Experimental evaluation is solid. Would love to see additional data regarding sensitivity to method length and robustness under changes to method names and signature.

Clarity: Paper is very well written.

Relation to Prior Work: Yes

Reproducibility: Yes

Additional Feedback: - Would have been great to see more discussion on what are the challenges of translation for programming languages vs. natural languages. What challenges are unique to source code? Just off the top of my head: + APIs and libraries + variable names + compositionality + purity / side effects + scoping rules + types - The problem of programming language translation is rarely the syntax of the language itself, but rather the semantic differences between the languages, and even worse, between their underlying libraries. You briefly touch on that in line 49, and also in Figure 10 (and in line 279) of the supplementary material. I had hoped for a more elaborate discussion of this point. - Robustness to variable renaming is a great experiment (Figure 8), do you have a more systematic experiment on the robustness that you can report numbers for? - What is the sensitivity of the model to method names? Looks like name and signature of the method would be critical for the translation to be successful? - As in other cases, the quality of translation probably degrades as method length increases? Can you share any numbers on that? - What makes translation easier between certain pairs of languages and harder between other pairs? - Have you tried translation between more foreign languages? Say C++ and Ocaml? - Have you tried translation between more nuanced language pairs, such as Python2 and Python3? - most of the examples are small algorithmic computations where the implementation is both common and relatively isolated (for example, max, sum_elements, no_letters from Figure 7). I wonder how the approach would work for functions that do not perform a specific isolated tasks, but combine several operations (e.g., load CSV from file and save it to a database). - Passing unit tests is unfortunately not a great metric either, but I agree that it is much better than reference match. Maybe consider more structured diff that allows to isolate subtrees of the AST where translation went wrong. UPDATE: Thank you for the thoughtful author response. Please include the experiments mentioned in the response in the final version of the paper.


Review 4

Summary and Contributions: This paper proposes to leverage recent approach in unsupervised machine translation to train a fully unsupervised neural transcompiler. The experimental results show that the proposed model can translate functions between C++, JAVA and Python. The authors also build and release a test set as parallel functions.

Strengths: The paper shows the proposed approach outperforms rule-based commercial baselines by a significant margin. The test set the authors share will be useful for the future research The paper proposes a new method to translate functions from a programming language to another, based on monolingual source code

Weaknesses: This is a fully unsupervised translation model, but I have interests in the case where we have a few supervised examples. How much important

Correctness: In Table 2, the performance gets better as the beam size increases. Did you keep increasing the beam size? How much is performance changed?

Clarity: The paper is mostly clearly written. Did you conduct ablation study of denoising auto-encoding and back-translation effect on this dataset?

Relation to Prior Work: None.

Reproducibility: Yes

Additional Feedback: UPDATE: Thank you for providing the authors' feedback.

[Author Response · NeurIPS 2020]

We thank the reviewers for their insightful remarks and suggestions. We have addressed them in the new version of the
paper. We now reply to the comments and questions raised by the reviewers, starting with common questions.

**[R2] [R4] Ablation studies.** We conducted ablation studies and found that pretraining is critical: models trained from
scratch fail to generate correct code. The DAE task is critical to initialize the decoding process (otherwise the model
never understands it has to decode, and the back-translated sentences are too noisy to give a learning signal), but we
found that it is possible to stop it after a few thousand iterations without impacting the performance. It is likely that
using more powerful models like T5 would achieve a better performance. However, the back-translation step requires to
translate functions on-the-fly at each iteration and using larger models significantly slows down the training (much
more than on classification tasks). For instance, using a 24-layer decoder for generation would be too slow, but mixing
a large encoder with a small decoder may be an option. In the context of natural languages, BART is trained with extra
tasks such as span prediction or sentence permutation. These tasks could easily be adapted to programming languages
and would be a promising direction for future work.

**[R2] [R4] Supervision.** For the pairs of programming languages we consider, we did not find any parallel datasets
large enough to be used for training. Consequently, we are not able to make comparisons with supervised approaches.
However, we agree this could be very valuable and it could be done in future work on language pairs where parallel
datasets exist (e.g. CoffeScript ↔ JavaScript). Moreover, a large parallel dataset could be useful to improve the
pretraining: As shown in Lample and Conneau (2019), pretraining with both masked-language modeling and translation
language modeling (TLM) objectives leads to a better performances in natural languages as the TLM objective provides
high quality cross-lingual embeddings.

**[R2] [R3] Function length v.s. performance.** For C++ → Python, the length of functions in the test set varies from
16 to 430 tokens. We observe that the performance decreases when the length increases. For beam 10, the performance
on functions with less than 45 tokens is of 72% accuracy, 30% accuracy for functions between 100 and 120 tokens, and
10% accuracy for functions with more than 210 tokens. Thank you for suggesting this study. It is very interesting and
we will report the full table in the updated version of the paper.

**[R1] Notations (Beam / Beam top-1 / CA@N)** What we refer to as "Beam N - Top 1" is indeed what people refer to
as "Beam N" in machine translation. We agree with the reviewer that this is confusing and we will update the paper
with standard notations and the recommended "CA@N" instead of "Beam N", to highlight that we are using a different
metric and to be consistent with the machine translation terminology.

**[R1] Negative Broader Impact Discussion.** We believe that the main negative consequence of developments in
programming languages translation would be a reduced employability for experts in archaic programming languages.
It is true that relying on ML-generated code could make IT systems more fragile, especially if the output of the ML
system is not human-readable. Besides, programmers might have too much confidence in the translator and fail to spot
errors they would have not made without the ML system. We will give it more thoughts in the Broader Impact section.

**[R3] Robustness to method and variable names** We manually tried to fool the model by providing input functions
with inconsistent names, for instance by renaming a function called "factorial" to "fibonacci" (or something totally
unrelated with the content of the function), or renaming a string variable to "number". We observed that the model is
robust to these modifications, and that this do not impact the correctness of translations. Instead, TransCoder properly
adapts the output to be more consistent with input names and types (c.f. Figure 8 in the appendix).

**[R3] Challenges in programming languages translation.** We agree that the paper would benefit from more elabo-
rate discussions about issues and challenges in programming languages translation. Reacting appropriately when the
source language uses a library with no equivalent in the target language is notably difficult. We did not observe this
issue at test time because functions from GeeksForGeeks typically do not rely on external libraries (e.g. NumPy or
SciPy), but this is indeed a current limitation of the model.

**[R3] Language pair difficulty.** TransCoder performs well on language pairs that share many keywords used for
similar purposes (anchor points). Having a similar syntax helps, but TransCoder is also able to translate python-specific
syntax to C++ or Java. The performance is lower when translating from Python. An explanation comes from the type
inference, an additional difficult task that the model does not have in the other directions. As there are many common
keywords between C++ and OCaml, and between Python2 and Python3, we expect that TransCoder would also perform
well in these directions. The confusions between Python2 and Python3 because of the languages similarities should be
mitigated by the use of the language token during decoding. It would be interesting to check whether the syntactic
differences between C++ and a functional language would not be too much of a barrier.

**[R4] Effect of higher beam sizes.** We generated translations with a beam of size 50, but did not observe a very large
difference compared to Beam 25. Improvements remain the most important between Beam 1 and Beam 5.

[Meta-Review · NeurIPS 2020]

Reviewers agree that this paper is a significant advance in the problem of language translation. One lingering concern is with the positioning of the paper. In particular, the introduction needs to do a better job in recognizing that this paper focuses on small self-contained units of code. In order to be useful in a software engineering context, a translation tool would have to address a number of problems that are not addressed by this work, such as major differences in the design patterns used by APIs in different languages. Without a proper acknowledgment of the limitations of the approach early in the paper, this paper could make it difficult to publish follow-up work.